# Molecular Characterization of Wilson’s Disease in Liver Transplant Patients: A Five-Year Single-Center Experience in Iran

**DOI:** 10.3390/diagnostics15192504

**Published:** 2025-10-01

**Authors:** Zahra Beyzaei, Melika Majed, Seyed Mohsen Dehghani, Mohammad Hadi Imanieh, Ali Khazaee, Bita Geramizadeh, Ralf Weiskirchen

**Affiliations:** 1Transplant Research Center, Shiraz University of Medical Sciences, Shiraz 7193711351, Iran; z.beyzaei@gmail.com (Z.B.); melika.majed76@gmail.com (M.M.); alikhazaeinejad@gmail.com (A.K.); 2Gastroenterology and Hepatology Research Center, Shiraz University of Medical Sciences, Shiraz 7193711351, Iran; dehghanism@gmail.com (S.M.D.); imaniehm@sums.ac.ir (M.H.I.); 3Department of Pathology, Medical School of Shiraz University, Shiraz University of Medical Sciences, Shiraz 7193711351, Iran; 4Institute of Molecular Pathobiochemistry, Experimental Gene Therapy and Clinical Chemistry (IFMPEGKC) RWTH University Hospital Aachen, D-52074 Aachen, Germany

**Keywords:** Wilson’s disease, copper, mutation analysis, ATP7B, genetic prevalence, consanguinity, liver transplantation, genetic heterogeneity, AlphaFold modeling

## Abstract

**Background/Objectives**: Wilson’s disease (WD) is an autosomal recessive disorder characterized by pathological copper accumulation, primarily in the liver and brain. Severe hepatic involvement can be effectively treated with liver transplantation (LT). Geographic variation in *ATP7B* mutations suggests the presence of regional patterns that may impact disease presentation and management. This study aims to investigate the genetic basis of WD in patients from a major LT center in Iran. **Methods**: A retrospective analysis was conducted on clinical, biochemical, and pathological data from patients suspected of WD who underwent evaluation for LT between May 2020 and June 2025 at Shiraz University of Medical Sciences. Genetic testing was carried out on 20 patients at the Shiraz Transplant Research Center (STRC). Direct mutation analysis of *ATP7B* was performed for all patients, and the results correlated with clinical and demographic information. **Results**: In total, 20 WD patients who underwent liver transplantation (15 males, 5 females) carried 25 pathogenic or likely pathogenic *ATP7B* variants, 21 of which were previously unreported. Fifteen patients were homozygous, and five were compound-heterozygous; all heterozygous combinations occurred in the offspring of second-degree consanguineous unions. Recurrent changes included p.L549V, p.V872E, and p.P992S/L, while two nonsense variants (p.E1293X, p.R1319X) predicted truncated proteins. Variants were distributed across copper-binding, transmembrane, phosphorylation, and ATP-binding domains, and in silico AlphaMissense scores indicate damaging effects for most novel substitutions. Post-LT follow-up showed biochemical normalization in the majority of recipients, with five deaths recorded during the study period. **Conclusions**: This single-center Iranian study reveals a highly heterogeneous *ATP7B* mutational landscape with a large proportion of novel population-specific variants and underscores the benefit of comprehensive gene sequencing for timely WD diagnosis and family counseling, particularly in regions with prevalent consanguinity.

## 1. Introduction

Wilson’s Disease (WD, OMIM 277900) is a rare autosomal recessive disorder of the copper metabolism caused by pathogenic variants in the *ATP7B* gene, located on chromosome 13q14.3 [1]. The *ATP7B* gene encodes a copper-transporting P-type ATPase predominantly expressed in the liver, with lower expression in the kidney, placenta, mammary glands, brain, and lungs. This transmembrane protein contains six metal-binding domains and eight transmembrane segments, forming a channel that mediates ATP-dependent copper transport across cellular membranes [2,3,4].

In hepatocytes, ATP7B is synthesized in the endoplasmic reticulum and subsequently trafficked to the trans-Golgi network (TGN). Under physiological conditions, ATP7B incorporates copper into apo-ceruloplasmin and facilitates biliary copper excretion. Loss-of-function mutations disrupt these processes, leading to the accelerated degradation of apo-ceruloplasmin and impaired biliary clearance of copper. As a result, copper accumulates in the liver, brain, cornea, kidneys, and other organs, with the hepatic and neurological systems being most severely affected [5,6].

Clinically, WD presents in two major phenotypes: hepatic and neurological forms. In pediatric patients, the hepatic form is more prevalent, ranging from asymptomatic liver enzyme elevations to overt hepatocellular dysfunction. Histopathological features include steatosis, chronic hepatitis, and variable fibrosis, which may progress to cirrhosis or, in severe cases, acute liver failure. In adult patients, neurological and psychiatric manifestations are more common, such as dysarthria, tremors, muscular rigidity, dystonia, anxiety, and personality changes. These symptoms can occur with or without concurrent hepatic involvement. Recent studies have highlighted age-dependent differences in the clinical spectrum, underscoring the importance of early recognition tailored to patient age [7,8,9,10,11].

The *ATP7B* gene spans approximately 80 kb of genomic DNA and comprises 21 exons (coding region ~4.3 kb) and 20 introns. To date, over 800 distinct mutations have been identified and cataloged in the Human Gene Mutation Database (HGMD) [12]. A detailed understanding of the mutational spectrum is essential for accurate molecular diagnosis and genotype–phenotype correlation in WD [13,14,15].

WD is a rare autosomal recessive disorder of the copper metabolism, with global prevalence estimates ranging from 1 in 30,000 to 1 in 100,000 individuals. However, in populations with high rates of consanguineous marriage, the incidence of WD is significantly elevated. In Iran, particularly in the southern regions, consanguineous unions are common, with approximately 38.6% of the population engaging in such marriages and a mean inbreeding coefficient (α) of 0.018. This likely contributes to a higher incidence of autosomal recessive disorders, including WD [16]. Consanguinity increases the likelihood of homozygosity for recessive mutations, thereby elevating the risk of developing WD. Consequently, the prevalence of WD in Iran is higher than the global average, highlighting the importance of genetic screening and early diagnosis in these populations.

Molecular genetic testing through direct mutation analysis is highly effective in confirming WD in symptomatic individuals and is invaluable for identifying at-risk family members, including asymptomatic carriers [17]. Current diagnostic approaches adhere to established guidelines, such as those from ESPGHAN and the updated AASLD Practice Guidance, which provide standardized criteria for the clinical, biochemical, and genetic evaluation of patients [7,18].

For patients suspected of having WD, the initial evaluation should include serum ceruloplasmin concentration and 24 h urinary copper excretion, which are first-line non-invasive diagnostic tests. Targeted *ATP7B* gene analysis enables the detection of many pre-symptomatic patients, reducing the need for invasive procedures [19,20]. Around 40% of pre-symptomatic individuals excrete less than 100 µg of copper per day, with significant increases in urinary copper typically only observed in later stages of the disease. Additionally, hepatic copper quantification alone is insufficient to rule out WD, emphasizing the necessity for robust diagnostic criteria [21,22].

These findings underscore the clinical significance of *ATP7B* sequencing before a liver biopsy, as genetic confirmation can help to avoid unnecessary procedures and reduce the diagnostic uncertainty arising from false-positive histological or biochemical results [23]. Liver biopsy should be reserved for cases where non-invasive testing is inconclusive. Regional testing has revealed multiple variants, some of which being unique to certain populations, highlighting the importance of geographic differences in understanding the functional impact of *ATP7B* [24,25].

In this study, we conducted genetic analysis on twenty WD patients and their family members, focusing on *ATP7B* variants identified in liver-transplanted (LT) patients and evaluating the predicted structural and functional effects of these mutations on the ATP7B protein.

## 2. Materials and Methods

### 2.1. Cohort and Ethics

Between May 2020 and June 2025, genetic analysis was performed on 20 patients who were clinically and biochemically suspected to have WD. They were referred for genetic analysis to the Shiraz Transplant Research Center (STRC) and Namazi Hospital in Shiraz, Iran. Clinical data, biochemical investigations, pathological data, and diagnostic imaging were retrospectively reviewed from both electronic and paper medical records. All patients in this cohort underwent LT. Notably, a subset of patients were offspring of consanguineous marriages.

Written informed consent was obtained from the patients’ parents or guardians prior to inclusion. This study was approved by the Ethics Committee of Shiraz University of Medical Sciences (Approval #: IR.SUMS.REC.1402.140. Approval date: 18 June 2023.) and conducted in accordance with the Declaration of Helsinki.

### 2.2. Clinical Diagnosis

The initial diagnosis of WD should be based on a combination of ophthalmologic, neurological, and hepatic findings. Kayser–Fleischer rings detected by slit-lamp examination indicate copper deposition in the cornea. Neurological manifestations include tremors, dystonia, ataxia, or other movement disorders. Hepatic involvement can present as acute liver injury, decompensated cirrhosis, or fulminant liver failure.

### 2.3. Biochemical Diagnosis

Conventional biochemical markers, such as low serum ceruloplasmin levels (<20 mg/dL) and elevated baseline 24 h urinary copper excretion (>100 µg/24 h), were also used as diagnostic criteria. Evaluation of renal function and nutritional status included measurements of blood urea nitrogen (BUN), serum creatinine (CREA), and serum albumin (ALB). Liver biopsy was performed for all patients to confirm the diagnosis. Subsequently, all of the affected and asymptomatic individuals underwent a thorough clinical reassessment, including liver biochemical profiling and genetic analysis.

### 2.4. Mutation Screening in ATP7B

Whole blood samples (5 mL) were collected from WD patients, as well as from their parents when applicable. The buffy coat was separated and stored at −20 °C for subsequent analysis. Genomic DNA was extracted from these samples using the standard protocol of the DNA Extraction Kit (Qiagen).

All coding exons and flanking intronic regions of the ATP7B gene were amplified by polymerase chain reaction (PCR) using genomic DNA from patients. Primers were designed with Oligo7 software to flank each of the 21 coding exons (primer sequences and conditions available upon request). PCRs were carried out in a 25 μL total volume containing 12.5 μL Master Mix Red (Ampliqon), 10 μM of each primer, and 10 ng of genomic DNA. The thermocycling protocol consisted of an initial denaturation at 95 °C for 10 min, followed by 29 cycles of 94 °C for 60 s (denaturation), 59–63 °C for 60 s (annealing), and 72 °C for 60 s (extension), with a final extension at 72 °C for 10 min.

Bi-directional sequencing of purified PCR products was performed using an ABI Prism 3500 Genetic Analyzer (Applied Biosystems). Sequence data were analyzed with the Mutation Surveyor software package (SoftGenetics) and compared against the ATP7B reference sequence (GenBank accession no. NM_000053.4). All detected variants were cross-referenced with the Genome Aggregation Database (gnomAD), Human Gene Mutation Database (HGMD), dbSNP, MutationTaster, SIFT, ClinVar, WilsonGen, Iranome database and relevant literature [26]. Novel variants were classified following the American College of Medical Genetics and Genomics (ACMG) guidelines [27,28].

### 2.5. Protein Modeling

The three-dimensional (3D) structure of the human wild-type ATP7B protein (P35670) was retrieved from the AlphaFold Protein Structure Database (entry AF-P35670-F1-v4) [29]. This structure was analyzed to identify functional domains, assess potential mutation impacts, and gain insights into the structural basis of ATP7B protein functionality in WD. The confidence of the structure at each residue is represented on a scale of 0–100, known as pLDDT, which corresponds to the model’s predicted score on the lDDT-Cα metric [30]. Missense variant effect predictions were conducted using AlphaMissense [31].

## 3. Results

### 3.1. Demographic and Clinical Data

Out of the 53 patients analyzed, 20 were found to carry mutations associated with WD. The cohort consisted of 15 males and 5 females, with an average age of 27.1 ± 0.6 years (Table 1). The average age at the time of transplantation was 21.2 ± 4.5 years. Eight patients (P2, P4, P6, P10, P16, P17, P19, and P20) were born to consanguineous parents, all of whom were second-degree relatives.

Donor types included both deceased donors (DD) and living donors (LD), with 14 patients receiving whole-organ grafts and 6 patients receiving partial grafts. Among living donors, familial relationships were noted, including parents (father or mother) in four cases. The majority of transplantations utilized the piggyback technique (16/20), while the remaining subjects were operated by standard procedures. Blood group compatibility between donors and recipients was carefully matched to reduce immunologic complications.

Post-transplant follow-up data (mean duration 3.5 ± 0.6 years) showed normalization or improvements in liver function tests in most patients. The majority of recipients remain alive, with only five reported deaths during the study period. Post-transplant biochemical parameters (ALT, AST, BUN, creatinine, and albumin) generally indicated favorable graft function and stable renal profiles. The Pediatric End-Stage Liver Disease (PELD) and Model for End-Stage Liver Disease (MELD) scores ranged widely, reflecting the heterogeneity of disease severity at transplantation.

### 3.2. Biochemical Results

As shown in Table 2, prior to transplantation, clinical evaluations revealed classical WD symptoms, including cirrhosis (confirmed in six patients) and elevated biochemical markers indicating hepatic dysfunction. Baseline serum ceruloplasmin levels ranged from 0.08 to 2.9 g/L, with elevated liver enzymes (alanine aminotransferase (ALT) and aspartate aminotransferase (AST)) observed. ALT values ranged from 20 to 189 U/L and AST values ranged from 20 to 977 U/L. Baseline 24 h urinary copper excretion varied significantly, with some patients showing levels as high as 520 µg/24 h. BUN levels ranged from 7 to 17 mg/dL, with most patients falling within the normal reference range, indicating preserved renal function in the majority of cases. Serum creatinine levels varied between 0.5 and 5.3 mg/dL, with one patient exhibiting elevated creatinine (5.3 mg/dL), suggesting impaired renal function, possibly related to disease severity or treatment effects. Serum albumin concentrations ranged from 2.0 to 4.9 g/dL, reflecting variations in liver synthetic function. While albumin is not a comprehensive measure of nutritional status, lower levels in some patients likely reflect a combination of advanced liver disease and associated malnutrition, which are common in WD prior to transplantation. Chelation therapy was administered to all patients before transplantation. The majority of patients (17/20; 85%) were treated with Penicillamine, while a smaller number (3/20; 15%) received Trientine. Additionally, Zinc was given to all patients.

### 3.3. Genomic Diagnostic Results

A total of 25 distinct variants were identified in the coding regions or adjacent splice sites of the *ATP7B* gene in 20 individuals, including 21 novel variants (Table 3). Among the patients, 15 had mutations in a homozygous state, while 5 showed compound heterozygosity. Of the variants found, 21 were previously unreported and 3 had been described in other populations. The new mutations include 2 nonsense and 23 missense variants (Figure 1 and Appendix A).

These variants were distributed across multiple functional domains of the *ATP7B* protein, including copper-binding domains, transmembrane domains, the putative ATP-binding domain, and ATP hinge regions. Chromosomal locations were referenced against the hg19 genome assembly, with nucleotide changes described according to transcript NM_000053.4.

Pathogenicity assessments, based on public databases such as HGMD, ClinVar, WilsonGen, and gnomAD, revealed that some variants were previously characterized as pathogenic or likely pathogenic, while others were novel and classified as variants of uncertain significance (VUS). Several recurrent variants, including p.L549V, p.V872E, and p.P992S/L, were observed in unrelated patients.

Notably, all heterozygous variants were identified in patients from consanguineous marriages, highlighting the role of familial inheritance in this cohort. These findings emphasize the genetic heterogeneity of WD within the Iranian population and underscore the critical need for comprehensive genetic screening to enable accurate diagnosis and personalized therapeutic strategies.

### 3.4. Structure and Functional Implications of ATP7B (P-Type ATPase): In Silico Analysis

The AlphaFold protein prediction provides a very accurate estimate of the 3D structure of the human ATP7B protein, with a high model confidence level in most regions (Figure 2A and Appendix A). Predictions made by artificial intelligence indicate the regions in which mutations might likely be benign or pathogenic (Figure 2B).

In silico analyses predicted that many of the novel missense mutations were damaging, although some variants were still classified as benign or VUS, pending further functional studies (cf. Table 3). However, it is difficult to estimate the clinical impact of each missense mutation. The two nonsense mutations at positions 1293 (E→X) and 1319 (R→X) clearly result in shortened ATP7B proteins. Predictions about missense mutations are more complex. For example, amino acid 633 is located in a flexible domain of ATP7B (Figure 3A), and the change from glutamine (Q) to proline (P) might induce a kink that alters the neighboring β-sheet. However, other amino acid substitutions are located within regions that are likely pathogenic. A change at position 998 from glycine (G) to cysteine (C) could impact the arrangement of the α-helical fold in that region of the ATP7B protein (Figure 3B). Similarly, the substitutions of amino acids 992 (P→L) (Figure 3C) and 1321 (I→T) (Figure 3D) might alter the conformation of the α-helical structure. It is evident that exchanging non-polar amino acids for charged amino acids will significantly affect the overall folding of the ATP7B protein.

Although not experimentally addressed, the substitution of the nonpolar amino acid valine at amino acid position 731 with the charged amino acid glutamic acid will alter the net charge in that region. This change may impact the folding not only at the affected position, but also in more distant regions (Figure 4).

## 4. Discussion

The molecular diagnosis of WD is crucial for patient management, as it helps identify at-risk family members and facilitates genetic counseling. It also guides therapeutic decisions, assessing prognosis, and prenatal diagnostic evaluations [36,37].

Clinically, most patients in our cohort with WD presented early in life, although the exact age of onset was often unclear. LT is typically recommended in WD patients who develop acute liver failure, especially when accompanied by Coombs-negative hemolytic anemia and rapid neurological deterioration. It is also considered in cases of decompensated cirrhosis or progressive hepatic dysfunction that does not respond to chelation therapy and supportive measures. In some cases, transplantation may be an option when severe neurological manifestations coincide with advanced liver disease and do not improve with medical therapy. Overall, LT not only corrects hepatic failure, but also restores normal copper metabolism by replacing the defective hepatic enzyme.

In addition to its established role in treating hepatic failure, LT has been reported in patients with WD who primarily present with neurological symptoms. According to a recent systematic review, LT can lead to the stabilization or improvement of neurological manifestations in some patients, although outcomes can vary [38]. While LT reliably corrects hepatic dysfunction and restores physiological copper metabolism, neurological progression may persist or even emerge post-transplant, highlighting the importance of careful patient selection and long-term neurological follow-up.

In our cohort, patients primarily displayed classical hepatic decompensation, with six already having biopsy-proven cirrhosis at the time of referral. This indicates a delayed diagnosis, despite known family history in some cases. Pre-LT chelation therapy with penicillamine or trientine, along with zinc supplementation, stabilized most patients until receiving a liver graft, following international recommendations. Post-operatively, liver enzymes, urinary copper, and albumin levels largely normalized, demonstrating that orthotopic grafts restore physiological copper export, regardless of genotype [39,40]. However, five post-transplantation fatalities indicate that LT does not eliminate perioperative risk or chronic complications. Notably, three deceased patients carried TM-region ATP7B variants, raising concerns about ongoing extrahepatic copper accumulation or neurological progression, despite successful LT. Evidence from other studies also suggests that neurological symptoms may persist or emerge after LT, emphasizing the need for long-term monitoring [41,42].

Genetically, twenty-one previously unreported mutations were identified, including nineteen missense variants (Q633P, V750I, F771V, D1446Y, P992S, G998C, L549V, V872E, Q228R, I1321T, V731E, E1082A, R198G, S254R, G1347V, M266L, and E127K) and two nonsense mutations (E1293X, R1319X). The nonsense variants are considered pathogenic, as they introduce premature stop codons that disrupt the reading frame and truncate the protein. Structurally, the mutations cluster in functionally critical motifs. Missense changes such as p.G998C, p.P992L/S, and p.I1321T localize within or near the pCu and pATP motifs essential for copper translocation and energy coupling. AlphaFold modeling suggests local helical destabilization likely impair enzymatic activity, while the nonsense variants truncate the protein before completion of the TM7-TM8 bundle, almost certainly resulting in loss-of-function alleles. Homozygous carriers of these mutations are predicted to require lifelong copper chelation therapy or LT.

Our five-year analysis highlights significant allelic diversity of *ATP7B* in Iranian liver-transplanted WD patients, with 84% (21/25) of detected changes being novel to public databases. This rate exceeds the global average of approximately 40% novel findings reported in multi-national studies, suggesting a unique regional spectrum likely influenced by founder events and endogamy in Southern Iran. Consanguinity was a significant factor, as all compound-heterozygous cases originated from second-degree related parents, consistent with high autozygosity reported in other Middle-Eastern WD series [15,43,44,45].

From a translational perspective, our findings support mandatory *ATP7B* sequencing in all suspected WD cases prior to liver biopsy, which can reduce invasive procedures and enable cascade testing in families, where 50% of first-degree relatives are obligate carriers. The identification of recurrent substitutions (e.g., p.L549V, p.V872E) supports the development of cost-effective, population-specific screening panels, while reporting novel variants to databases like ClinVar refines global pathogenicity curation. Coupling genomic data with AlphaMissense scoring provides rapid functional insight, although confirmation via cell-based copper-transport assays remains necessary for variants of uncertain significance. Overall, this approach enhances diagnostic accuracy and guides personalized management strategies for WD patients, aligning our results with observations from other national and international cohorts [46,47].

## 5. Study Limitations

This retrospective study was conducted at a single center and included 20 patients with WD who had undergone LT. This study may have a selection bias towards patients with severe hepatic phenotypes, potentially resulting in an underrepresentation of cases primarily presenting with neurological symptoms. Additionally, brain MRI data and standardized assessments of neurological disease severity were not available, limiting the evaluation of extrahepatic manifestations. The functional validation of novel variants was restricted to in silico predictions and was not supported by biochemical assays. Furthermore, long-term follow-up beyond five years post-LT was not accessible, which hinders the ability to draw prognostic conclusions. Similarly to previous reports, a complete mutation detection rate was not achievable. In our cohort, 37.7% of ATP7B variants remained unidentified, possibly due to deep intronic changes affecting splicing or large deletions/rearrangements that were not detectable with our methodology.

## 6. Conclusions

This five-year survey of Iranian liver-transplanted WD patients revealed 25 *ATP7B* variants, 21 of which were previously unknown, highlighting a diverse regional mutation spectrum. Comprehensive sequencing played a crucial role in quick diagnosis and providing family counseling in a population with high rates of consanguinity. This study also identified recurring mutations (such as p.L549V, p.V872E, and p.P992S/L) that could be used to develop targeted screening panels. These findings contribute to global variant databases and emphasize the need for further research to understand how each new mutation affects ATP7B structure, copper transport, and response to treatment.

## Figures and Tables

**Figure 1 diagnostics-15-02504-f001:**
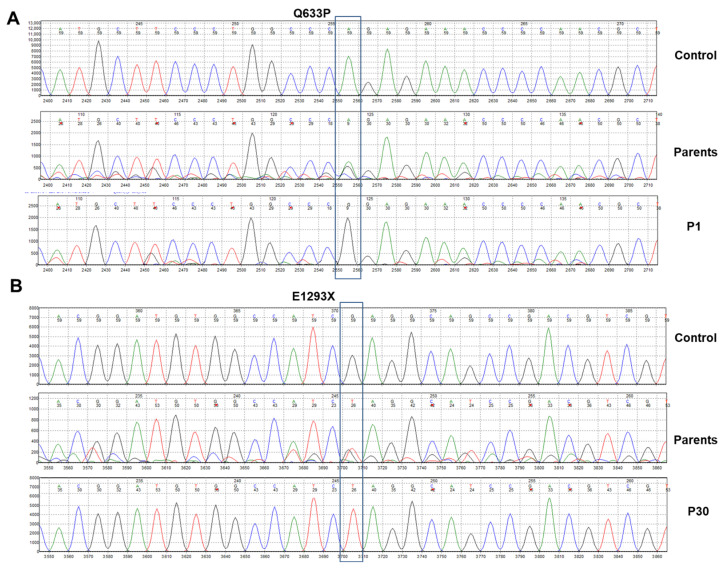
DNA sequence analysis from two representative cases. (**A**) Sequencing results for patient 1 (P1) affected by an amino acid substitution at position 633, resulting in the ATP7B missense variant Q633P are depicted. The upper panel shows the sequencing result at this location from a healthy individual (control). The middle panel shows the sequence result from one of the two parents who carry the mutation in a heterozygous fashion (parents). (**B**) Sequencing results for patient 30 (P30) affected by a nonsense mutation at position 1293 are depicted. The upper panel shows the sequencing result at this location from a healthy individual. The middle panel shows the sequence result from one of the two parents who carry the mutation in a heterozygous fashion.

**Figure 2 diagnostics-15-02504-f002:**
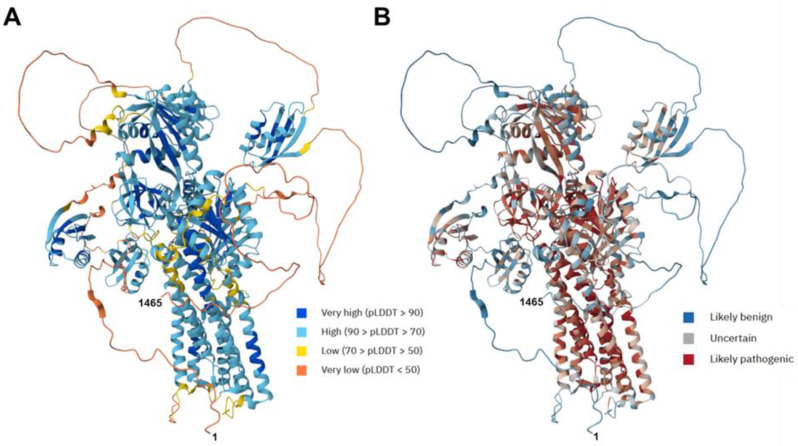
Predicted structure of human ATP7B protein and the predicted impact of novel ATP7B gene mutation identified in this study. (**A**) Structure prediction of the human ATP7B protein. The model was generated using AlphaFold. The per-residue estimate of the model confidence is given on a pLDDT scale ranging from 0 to 100, and is displayed in a color code indicating very low (pLDDT < 50) to very high (pLDDT > 90) estimates for the structure. (**B**) The pathogenicity of missense variants in the human ATP7B protein was estimated using AlphaMissense. The displayed color for each residue is the average AlphaMissense pathogenicity score across all possible amino acid substitutions at that position. For orientation, the location of the N-terminal (1) and C-terminal (1465) amino acids are marked.

**Figure 3 diagnostics-15-02504-f003:**
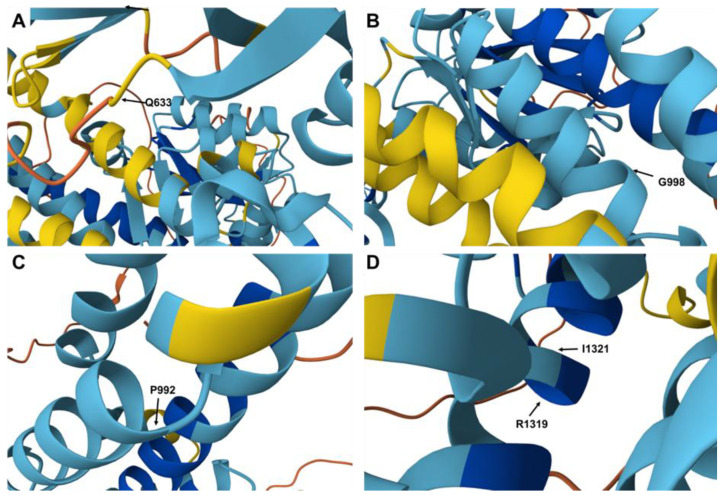
Location of specific amino acids in the human ATP7B protein. (**A**–**D**) Amino acids Q633, G998, P992, R1319, and I1321 are highlighted. Mutations at these sites could cause significant changes in the spatial folding of the ATP7B protein. In particular, a mutation changing R1319 to a stop codon would result in a much shorter ATP7B protein missing a large portion of the C-terminus.

**Figure 4 diagnostics-15-02504-f004:**
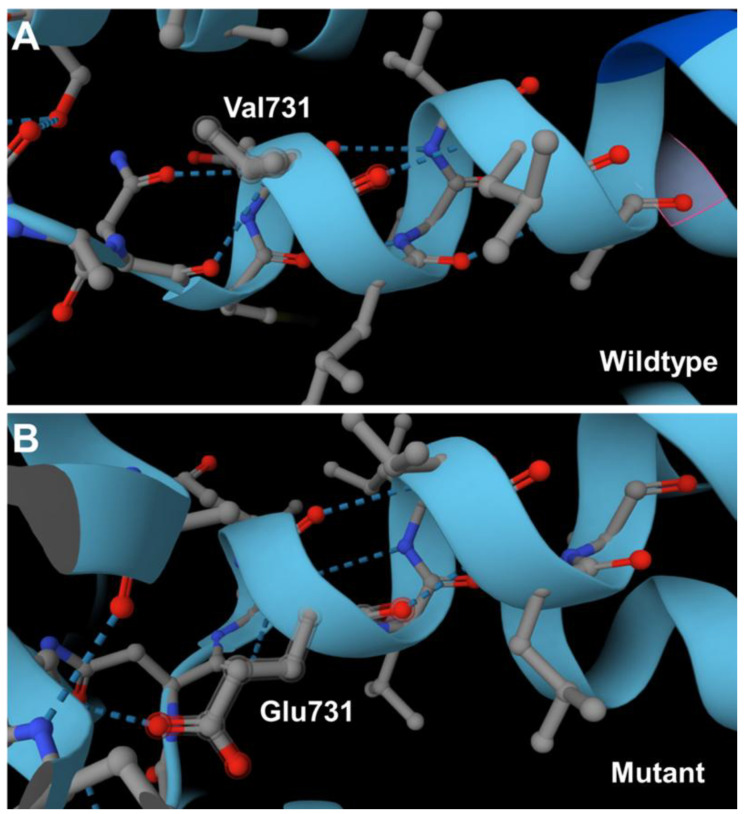
Potential impact of the amino acid substitution at position 731 in the human ATP7B protein. (**A**) According to fold prediction the amino acid valine 731 is located at the end of an alpha-helical segment of the human ATP7B protein. (**B**) The substitution of valine 731 with glutamic acid introduces new negative charges that have the potential to modify the structure by creating new hydrogen bonds. This could potentially lead to alterations in the overall protein fold. The negative charge can establish new hydrogen bonds with adjacent and distant protein regions.

**Table 1 diagnostics-15-02504-t001:** Patient’s characteristics before and after liver transplantation.

Variable	Overall Patients (*n* = 20)
Gender (M/F)	13/7
Current Mean age (years)	27.1 ± 0.6
Mean age at transplantation (years)	21.2 ± 4.5
Parental consanguinity *n* (%)	8 (40)
Type of donor, *n* (%)	
Deceased donor (whole)	14 (70%)
Living donor (partial)	6 (30%)
Outcome	
Alive	12 (60%)
Dead	8 (40%)
Main clinical presentations, *n*	
Hepatological	Liver cirrhosis 9, Jaundice 5, Acute liver injury 2
Neurological	Tremor 2, Ataxia 2
Psychiatric	-
Chelation therapy, *n* (%)	
Penicillamine	17 (85)
Trientine	3 (15)
Follow-up duration (years)	3.5 ± 0.6

**Table 2 diagnostics-15-02504-t002:** Laboratory data before and after liver transplantation (*n* = 20).

Parameter	Pre-Transplant (Mean ± SD)	Post-Transplant (Mean ± SD)
ALT (U/L)	90 ± 65	32 ± 21
AST (U/L)	95 ± 70	28 ± 18
BUN (mg/dL)	12 ± 3	13 ± 4
CREA (mg/dL)	0.9 ± 0.5	1.0 ± 0.4
ALB (g/dL)	3.2 ± 0.9	4.2 ± 0.4
SCER	0.49 ± 0.80	-
Urine copper (µg/24 h)	350 ± 200	–

Values are presented as mean ± standard deviation or number (%). Abbreviations: ALB, albumin; ALT, alanine aminotransaminase; AST, aspartate aminotransferase; BUN, Blood Urea Nitrogen; CREA, creatinine; SCER, serum ceruloplasmin.

**Table 3 diagnostics-15-02504-t003:** Mutation analysis of Wilson patients.

Patients/ Gender	Region of Protein *	Chr: Loc (hg19)	Exome/ Intron	Nucleotide Change	Protein Change	Variant Type	Zygosity	Previous Definition and Pathogenicity in HGMD	WilsonGen [32]	ClinVar [33]	gnomAD (Allele Frequency) [34]
P1/M	-	13:52536021	E5	c.1898A > C	p.Q633P	Missense	Homozygous	ND, P, Da	ND	ND	ND
P2/M	- TM4	13:52532554 13:52532491	E8 E8	c.2248G > A c.2311T > G	p.V750I p.F771V	Missense Missense	Heterozygous Heterozygous	ND, B, Da ND, P, Da	ND ND	ND ND	ND ND
P4/F	COOH	13:52508954	E21	c.4336G > T	p.D1446Y	Missense	Homozygous	ND, B	ND	ND	ND
P5/M	TM6 TM6	13:52520506 13:52520488	E13 E13	c.2974C > T c.2992G > T	p.P992S p.G998C	Missense Missense	Heterozygous Heterozygous	ND, LP, Da ND, LP, Da	ND ND	ND ND	ND ND
P16/M	-	13:52524464	E10	c.2519C > T	p.P840L	Missense	Homozygous	D, P, Da	P	P	P, 6.20 × 10^−7^
P18/M	HMA5 -	13:52542642 13:52524258	E5 E11	c.1645C > G c.2615T > A	p.L549V p.V872E	Missense Missense	Heterozygous Heterozygous	D, P, Da ND, P, Da	ND ND	ND ND	ND ND
P23/M	-	13:52548673	E2	c.683A > G	p.Q228R	Missense	Homozygous	ND, B	ND	ND	ND
P24/M	HMA5	13:52542642	E5	c.1645C > G	p.L549V	Missense	Homozygous	D, P, Da	ND	ND	ND
P30/F	-	13:52511638	E18	c.3877G > T	p.E1293X	Nonsense	Homozygous	ND, P, Da	ND	ND	ND
P31/M	- TM7	13:52524258 13:52511471	E11 E19	c.2614G > A c.3962T > G	p.V872E p.I1321T	Missense Missense	Heterozygous Heterozygous	ND, P, Da ND, P, Da	ND ND	ND ND	ND ND
P32/F	TM3	13:52532610	E8	c.2192T > A	p.V731E	Missense	Homozygous	D, LP, Da	ND	ND	ND
P33/M	-	13:52516689	E14	c.3245A > C	p.E1082A	Missense	Homozygous	ND, P, Da	ND	ND	ND
P36/F	-	13:52548763	E2	c.593G > A	p.R198G	Missense	Homozygous	D, P, Da	ND	ND	ND
P37/M	-	13:52524258	E11	c.2614T > A	p.V872E	Missense	Homozygous	ND, P, Da	ND	ND	ND
P39/F	-	13:52548596	E2	c.760A > C	p.S254R	Missense	Homozygous	ND, B	ND	ND	ND
P40/F	-	13:51937342	E19	c.3955C > T	p.R1319X	Nonsense	Homozygous	D, P, Da	P	P	P, 1.40 × 10^−4^
P41/M	- HMA3	13:52509813 13:52548560	E19 E2	c.4040G > T c.953A > C	p.G1347V p.M266L	Missense Missense	Heterozygous Heterozygous	ND, P, Da ND, P, Da	ND ND	ND ND	ND ND
P45/M	-	13:52532542	E8	c.2260G > A	p.E754K	Missense	Homozygous	D, P, Da	VUS	VUS	VUS
P49/M	TM6	13:52520505	E13	c.2975C > T	p.P992L	Missense	Homozygous	D, P, Da	LP	LB	LB, 6.20 × 10^−7^
P52/M	-	13:52548977	E2	c.379G > A	p. E127K	Missense	Homozygous	ND, B	ND	ND	ND

***** The information was extracted from accession number NM_000053.4 deposited in GenBank [35] (refer to Appendix A for more details). Novel unclassified mutations are highlighted in gray shade. Abbreviations include the following: B, benign; COOH, C-terminus; D, defined; Da, damaging; LB, likely benign; LP, likely pathogenic; ND, not defined; P, pathogenic; TM, transmembrane domain; VUS, variant of uncertain significance.

## Data Availability

The data generated for this study are available upon reasonable request from the corresponding author B.G.

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
