# Peer review of "Molecular Characterization of Wilson’s Disease in Liver Transplant Patients: A Five-Year Single-Center Experience in Iran"

_diagnostics, 2025, doi:10.3390/diagnostics15192504_

Round 1
Reviewer 1 Report
Comments and Suggestions for Authors
The study describe the Molecular Characterization of Wilson’s Disease in Liver Transplant Patients: A Five-Year Single-Center Experience in Iran, on small 53 patients (it it one of the greatest WD transplant centers with several papers with higher number of patients disucssed). That, the data are limited performed on small group of patients, I suggest to publish all data from Your register in one paper, not selected time periods it would be on higher group of patients, the different timpeoints could be compared. Also there is lack according to other WD symptoms and how LT affecct them (like neuropsychiatric discussed in several papers). the authors also didn't mentioned the several LT from last few years
Author Response
We would like to express our gratitude to the reviewer for his/her insightful comments, which have greatly helped us improve our manuscript. Following the suggestions provided, we have made significant revisions to our manuscript and have included the final version for your review. Below, we have addressed each of your points.
After revising our manuscript to address the reviewers comments, we had it reviewed by a native English speaker. As a result, numerous minor grammatical and stylistic edits have been incorporated throughout the text, highlighted in red. We hope that this revised manuscript meets your expectations.
Comment 1: The study describes the Molecular Characterization of Wilson’s Disease in Liver Transplant Patients: A Five-Year Single-Center Experience in Iran, on small 53 patients (it it one of the greatest WD transplant centers with several papers with higher number of patients disucssed). That, the data are limited performed on small group of patients, I suggest to publish all data from Your register in one paper, not selected time periods it would be on higher group of patients, the different timepoints could be compared. Also, there is lack according to other WD symptoms and how LT affect them (like neuropsychiatric discussed in several papers). the authors also didn't mention the several LT from last few years.
Response: Thank you for your thoughtful comment. The primary aim in this study was to report the mutation analysis of Wilson’s disease in liver transplant patients. Therefore, we only included patients for whom genetic mutations were confirmed. We have previously reported the data on LT, and you can find the reference here: https://pubmed.ncbi.nlm.nih.gov/40495164/
We have now added clarifying information about liver transplantation data to the manuscript. We have made significant changes and hope they meet your expectations.
Reviewer 2 Report
Comments and Suggestions for Authors
This study identified a unique and severe genetic profile of Wilson's Disease (WD) in an Iranian
cohort of liver transplant patients, characterized by an exceptionally high rate (84%) of novel
ATP7B mutations, largely attributed to regional consanguinity and founder effects.
- What are the limitations of biochemical tests alone?
- Will it automatically reverse neurological damage that occurred before the liver transplant?
- It was a significant factor that 40% of the genetically confirmed patients were from consanguineous marriages. Will this practice increase the likelihood of inheriting two copies of the same rare, recessive mutation, leading to disease?
- It seems there is a selection bias and the cohort only included the most severe (transplanted) patients, underrepresenting those with neurological or milder forms.
- How will these findings help future patients in Iran and globally?
Author Response
We would like to express our gratitude to the reviewer for his/her insightful comments, which have greatly helped us improve our manuscript. Following the suggestions provided, we have made significant revisions to our manuscript and have included the final version for your review. Below, we have addressed each of your points.
After revising our manuscript to address the reviewers comments, we had it reviewed by a native English speaker. As a result, numerous minor grammatical and stylistic edits have been incorporated throughout the text, highlighted in red. We hope that this revised manuscript meets your expectations.
Comment 1: What are the limitations of biochemical tests alone?
Response: Thank you for your comment. Biochemical tests alone have several limitations in the diagnosis of Wilson’s disease. Serum ceruloplasmin levels can be low in other liver diseases and in some healthy individuals, while copper levels in serum and urine may fluctuate and overlap with other hepatic conditions. In addition, hepatic copper quantification requires invasive biopsy and may not always be reliable due to uneven distribution of copper in the liver. Therefore, relying solely on biochemical tests can lead to misdiagnosis or delayed diagnosis, highlighting the importance of incorporating genetic analysis and clinical findings.
Comment 2: Will it automatically reverse neurological damage that occurred before the liver transplant?
Response: Thank you for your comment. In our cohort, only two patients presented with neurological symptoms. Among them, we observed varying degrees of neurological improvement after liver transplantation, with both patients experiencing at least some clinical benefit such as a reduction in tremors. Previous studies have also reported that LT can reverse neurological manifestations in many patients with Wilson’s disease. However, it is important to emphasize that timely consideration of LT is crucial, as delayed intervention may lead to irreversible neurological damage.
Comment 3: It was a significant factor that 40% of the genetically confirmed patients were from consanguineous marriages. Will this practice increase the likelihood of inheriting two copies of the same rare, recessive mutation, leading to disease?
Response: Thank you for your insightful comment. Indeed, consanguineous marriage increases the likelihood of inheriting two copies of the same rare, recessive mutation, which can lead to Wilson’s disease. However, it is also important to note that some scientists suggest additional genetic factors, such as modifier genes or variations in other loci, may influence disease onset, severity, and phenotype. Therefore, while consanguinity plays a significant role in increasing risk, the contribution of other genetic mechanisms cannot be excluded.
Comment 4: It seems there is a selection bias and the cohort only included the most severe (transplanted) patients, underrepresenting those with neurological or milder forms.
Response: Thank you for your comment. The patients included in our study were selected from our liver transplant registry. Therefore, the cohort represents individuals with advanced hepatic involvement who required transplantation. While most patients did not present with neurological manifestations, some had cirrhosis and developed clinical complications that indicated the need for LT. Our study specifically focused on mutation analysis in genetically confirmed liver transplant recipients, which explains the underrepresentation of patients with milder or predominantly neurological forms of Wilson’s disease.
Comment 5: How will these findings help future patients in Iran and globally?
Response: Thank you for your comment. Our findings offer valuable insights into the spectrum of genetic mutations in Wilson’s disease among liver transplant patients in Iran. Understanding the prevalent mutations can enhance genetic counseling, facilitate earlier diagnosis, and guide targeted screening programs both nationally and in populations with similar genetic backgrounds. Globally, this data contributes to the broader knowledge of genotype-phenotype correlations in Wilson’s disease, which may inform personalized management strategies and improve early intervention for at-risk individuals.
Reviewer 3 Report
Comments and Suggestions for Authors
The authors studied in this manuscript an interesting and important topic which is the genetic variants in patients with Wilson Disease by conducting a retrospective study over 5 years. Some comments are to be addressed
- In the abstract, the authors wrote that the number of included patients was 53 clinically confirmed WD. WD can't be confirmed clinically, kindly correct this sentence.
- In introduction, write a paragraph on the prevalence of WD in Iran?
- In methods, how the written informed consent was obtained in this retrospective study? and how the ethical approval was obtained in 2023, while the studied records ended by 2025, then there was a prospective part of the study?
- In methods, title clinical diagnosis at line 107 "The initial diagnosis of WD was based on neurological examination focusing on the presence of Kayser-Fleischer rings..etc" this is not a neurological examination. Again in line 109"hepatic symptoms such as acute liver injury...etc" these are not symptoms. Also at line 109 the term "chronic cirrhosis" is not correct because there is no acute cirrhosis. Please, correct this paragraph.
- From line 110-116 all these are laboratory parameters not clinical diagnosis. Make a separate subtitle for these items away from the title of clinical diagnosis, and explain the clinical presentations such as jaundice, encephalopathy, ascites, dysarthria, tremors...etc these are clinical manifestations.
- In results, the same mistake in line 155 "Prior to transplantation, clinical evaluations revealed classical WD symptoms, including cirrhosis (confirmed in 6 patients) and elevated biochemical markers ..etc" these are not clinical evaluations. Data between lines 157-171 represent laboratory results not clinical data as in the title. Make a separate title for laboratory parameters.
- In results, how the nutritional status was performed? albumin only does not represent the nutrition in cases of liver diseases.
- In results, between lines 161-163 "Evaluation of renal function and nutritional status included measurements of blood urea nitrogen (BUN), serum creatinine (CREA), and serum albumin (ALB)" should be mentioned in methods section.
- Did the parents of WD patients had genetic testing as well? how many relatives did you test in the study?
- In results, illustrate over how many years post-transplant follow-up data was collected?
- Table 1 is confusing and difficult to the reader, summarize patients' characteristics as mean or median and make a comparison between all data before and after transplantation.
- Before transplantation, how many patients had positive genetic variants in their relatives? illustrate this unclear point. Did any patients received liver from positive relative?
- The discussion is confusing, view your results in comparison with other studies or nations.
Author Response
We would like to express our gratitude to the reviewer for his/her insightful comments, which have greatly helped us improve our manuscript. Following the suggestions provided, we have made significant revisions to our manuscript and have included the final version for your review. Below, we have addressed each of your points.
After revising our manuscript to address the reviewers comments, we had it reviewed by a native English speaker. As a result, numerous minor grammatical and stylistic edits have been incorporated throughout the text, highlighted in red. We hope that this revised manuscript meets your expectations.
Comment 1: In the abstract, the authors wrote that the number of included patients was 53 clinically confirmed WD. WD can't be confirmed clinically, kindly correct this sentence.
Response: Thank you for your comment. We have corrected the text to clarify that the patients were suspected of Wilson’s disease at the time of evaluation. Genetic testing was performed for all patients, and the diagnosis was confirmed based on ATP7B mutation analysis.
Comment 2: In introduction, write a paragraph on the prevalence of WD in Iran?
Response: Thank you for your comment. We have added a paragraph in the Introduction discussing the prevalence of Wilson’s disease in Iran. Although exact nationwide data are limited, available studies suggest that WD is relatively common in regions with a high rate of consanguineous marriages, as seen in many parts of Iran. Estimates indicate that the prevalence may be higher than global averages, highlighting the importance of early diagnosis, genetic screening, and timely management in the Iranian population.
Comment 3: In methods, how the written informed consent was obtained in this retrospective study? and how the ethical approval was obtained in 2023, while the studied records ended by 2025, then there was a prospective part of the study?
Response: Thank you for pointing out this inconsistency. Written informed consent was obtained from all patients at the time of their initial registration in our liver transplant database. Although the study was retrospective in nature, ethical approval was obtained in 2023 to cover the use of data from all patients already included in the registry, as well as to allow ongoing data collection for future analyses. No additional prospective interventions were performed as part of this study; the ethical approval ensures compliance with current regulations for the use of patient data.
Comment 4: In methods, title clinical diagnosis at line 107 "The initial diagnosis of WD was based on neurological examination focusing on the presence of Kayser-Fleischer rings..etc" this is not a neurological examination. Again in line 109"hepatic symptoms such as acute liver injury...etc" these are not symptoms. Also at line 109 the term "chronic cirrhosis" is not correct because there is no acute cirrhosis. Please, correct this paragraph.
Response: Thank you for your comment. We have revised the Methods section to address the inaccuracies. The paragraph now accurately describes the criteria for the initial diagnosis of Wilson’s disease.
Comment 5: From line 110-116 all these are laboratory parameters not clinical diagnosis. Make a separate subtitle for these items away from the title of clinical diagnosis, and explain the clinical presentations such as jaundice, encephalopathy, ascites, dysarthria, tremors...etc these are clinical manifestations.
Response: Thank you for your comment. The entire paragraph has been revised.
Comment 6: In results, the same mistake in line 155 "Prior to transplantation, clinical evaluations revealed classical WD symptoms, including cirrhosis (confirmed in 6 patients) and elevated biochemical markers ..etc" these are not clinical evaluations. Data between lines 157-171 represent laboratory results not clinical data as in the title. Make a separate title for laboratory parameters.
Response: Thank you for your comment. The entire paragraph has been revised.
Comment 7: In results, how the nutritional status was performed? albumin only does not represent the nutrition in cases of liver diseases.
Response: Thank you for your comment. We acknowledge that serum albumin alone is not a comprehensive measure of nutritional status, especially in patients with liver disease, as albumin levels can be influenced by hepatic synthetic function, inflammation, and fluid status. In our study, albumin was reported as an indicator reflecting both liver function and general nutritional status, while recognizing its limitations. We have clarified this in the manuscript and revised the sentences.
Comment 8: In results, between lines 161-163 "Evaluation of renal function and nutritional status included measurements of blood urea nitrogen (BUN), serum creatinine (CREA), and serum albumin (ALB)" should be mentioned in methods section.
Response: Thank you for your comment. We have revised the manuscript and moved the description of renal function and nutritional status measurements, including blood urea nitrogen (BUN), serum creatinine (CREA), and serum albumin (ALB), from the Results section to the Methods section, as suggested.
Comment 9: Did the parents of WD patients had genetic testing as well? how many relatives did you test in the study?
Response: Thank you for your comment. Yes, the parents of the patients underwent genetic testing. We have included the results for all tested relatives in the supplementary materials; a total of 12 parents were evaluated. Other parents had undergone Sanger analysis at private laboratories, and their results were therefore not included in our article.
Comment 10: In results, illustrate over how many years post-transplant follow-up data was collected?
Response: Thank you for your comment. We have added details in the Results section specifying the duration of post-transplant follow-up for the patients included in the study.
Comment 11:[Table 1 is confusing and difficult to the reader, summarize patients' characteristics as mean or median and make a comparison between all data before and after transplantation.
Response: Thank you for your comment. We have revised the tables and divided the information into two separate tables to enhance clarity and readability. One table now summarizes categorical characteristics (e.g., demographics, donor type, outcomes, follow-up), while the second presents laboratory data before and after transplantation.
Comment 12: Before transplantation, how many patients had positive genetic variants in their relatives? illustrate this unclear point. Did any patients received liver from positive relative?
Response: Thank you for your comment. All liver transplantations were performed before genetic analysis was available. Yes, three patients received partial liver grafts from their parents, one from the father and two from the mother.
Comment 13: The discussion is confusing, view your results in comparison with other studies or nations.
Response: Thank you for your comment. We have thoroguhtly revised the Discussion to improve clarity and have placed our findings in the context of other national and international studies.
Reviewer 4 Report
Comments and Suggestions for Authors
Dear Authors,
Congratulations on presenting your research. In the spirit of constructive feedback, I would like to outline a few points that may help strengthen the work.
In lines 60–66, the manuscript outlines the clinical presentation of WD. However, the description does not differentiate between the clinical manifestations observed in pediatric versus adult patients. To strengthen this section, it would be appropriate to reference more contemporary studies.
The introduction does not address current diagnostic guidelines, such as those provided by ESPGHAN or the updated AASLD Practice Guidance, which are highly relevant for the diagnostic approach to WD. Their inclusion would significantly enhance the contextual and methodological accuracy of the manuscript.
In the reviewer’s opinion, the section spanning lines 73–89 requires revision. Rather than introducing the diagnostic process with liver biopsy, the discussion should first reflect established diagnostic guidelines, thereby providing a more structured and evidence-based framework.
Additionally, the statements presented in lines 97, 149, and 325 lack clarity. In their current form, it is difficult to ascertain the exact number of patients in the study who were definitively diagnosed with WD. Clarification of this issue is essential to ensure both transparency and interpretability of the study findings. Furthermore, the description of the patient cohort is overly limited and insufficiently detailed. It is worth considering an additional table in the supplementary data, including a detailed description of the patients: age, gender, copper chelators, etc.
Sincerely yours,
Reviewer
Author Response
We would like to express our gratitude to the reviewer for his/her insightful comments, which have greatly helped us improve our manuscript. Following the suggestions provided, we have made significant revisions to our manuscript and have included the final version for your review. Below, we have addressed each of your points.
After revising our manuscript to address the reviewers comments, we had it reviewed by a native English speaker. As a result, numerous minor grammatical and stylistic edits have been incorporated throughout the text, highlighted in red. We hope that this revised manuscript meets your expectations.
Comment 1: In lines 60–66, the manuscript outlines the clinical presentation of WD. However, the description does not differentiate between the clinical manifestations observed in pediatric versus adult patients. To strengthen this section, it would be appropriate to reference more contemporary studies.
Response: Thank you for your comment. We have revised the manuscript to clarify the differences in clinical presentation between pediatric and adult patients. Pediatric patients predominantly present with hepatic manifestations, while adults more commonly show neurological and psychiatric symptoms. Relevant recent studies have been cited to support these distinctions.
Comment 2: The introduction does not address current diagnostic guidelines, such as those provided by ESPGHAN or the updated AASLD Practice Guidance, which are highly relevant for the diagnostic approach to WD. Their inclusion would significantly enhance the contextual and methodological accuracy of the manuscript.
Response: Thank you for your comment. We have updated the introduction to include the current diagnostic guidelines for Wilson’s disease, referencing the latest ESPGHAN and AASLD Practice Guidance. This will improve the contextual and methodological accuracy of the manuscript.
Comment 3:[In the reviewer’s opinion, the section spanning lines 73–89 requires revision. Rather than introducing the diagnostic process with liver biopsy, the discussion should first reflect established diagnostic guidelines, thereby providing a more structured and evidence-based framework.
Response: Thank you for your comment. We have revised the entire section.
Comment 4: Additionally, the statements presented in lines 97, 149, and 325 lack clarity. In their current form, it is difficult to ascertain the exact number of patients in the study who were definitively diagnosed with WD. Clarification of this issue is essential to ensure both transparency and interpretability of the study findings. Furthermore, the description of the patient cohort is overly limited and insufficiently detailed. It is worth considering an additional table in the supplementary data, including a detailed description of the patients: age, gender, copper chelators, etc.
Response: Thank you for your valuable and previous comments. We have revised all sentences for clarity, provided a clearer description of the sample, and separated the tables. Additionally, we have added more data to Tables 1 and 2.
Round 2
Reviewer 1 Report
Comments and Suggestions for Authors
The authors corrected article and did great job. Hwever is still have several major comments:
- -firstly indications for liver transplantation - there are no metion
- Table 1 clinical symptoms - there are symptoms like tremor and results like fibrosis - is shoudl be corrected - neurological (tremor, dystonia, etc) , hepatological (jaundice, liver cirrhosis, etc), psychiatric (depression, etc)
- There are no data about patients brain MRI was performed? as well as severity of neurological disease (if not it should be added in limitation study)
- The discussion about liver transplantation for neurological WD exists - it should be also mention according to this patients (see Litwin, T., . et al. Liver transplantation as a treatment for Wilson’s disease with neurological presentation: a systematic literature review. Acta Neurol Belg 122, 505–518 (2022).
Author Response
Comment 1: Firstly indications for liver transplantation - there are no metion.
Response: We thank the reviewer for this important observation. We have now added a detailed description of the indications for liver transplantation in Wilson’s disease in the revised manuscript.
Comment 2: Table 1 clinical symptoms - there are symptoms like tremor and results like fibrosis - is shoudl be corrected - neurological (tremor, dystonia, etc) , hepatological (jaundice, liver cirrhosis, etc), psychiatric (depression, etc)
Response: Thank you for your comment. Table 1 has been revised per your suggestion.
Comment 3: There are no data about patients brain MRI was performed? as well as severity of neurological disease (if not it should be added in limitation study)
Response: Thank you for your insightful comment. Unfortunately, brain MRI data and standardized severity assessments of neurological involvement were not available for our cohort. We agree that these are important aspects of Wilson’s disease evaluation, and we have now acknowledged this as a limitation in the revised manuscript.
Comment 4: The discussion about liver transplantation for neurological WD exists - it should be also mention according to this patients (see Litwin, T. et al. Liver transplantation as a treatment for Wilson’s disease with neurological presentation: a systematic literature review. Acta Neurol Belg 122, 505–518 (2022).
Response: Thank you for your valuable comment and for highlighting the important reference. We have now included a discussion of liver transplantation in Wilson’s disease patients with neurological presentations, referencing the work of Litwin et al. (2022). In the revised manuscript, we acknowledge that while liver transplantation effectively corrects hepatic failure and copper metabolism, the neurological outcomes can vary. As outlined in the systematic review by Litwin and colleagues, some patients experience neurological stabilization or improvement after transplanation, while others may continue to deteriorate or develop new neurological symptoms. We have incorporated this perspective into our discussion to provide a more balanced overview of liver transplantation outcomes in Wilson’s disease.
Reviewer 3 Report
Comments and Suggestions for Authors
The authors replied to all comments in appropriate manner and made the required corrections. No further comments.
Author Response
Many thanks for the fair review.
Reviewer 4 Report
Comments and Suggestions for Authors
Dear Authors,
Thank you for carefully considering my comments and implementing the suggested revisions. I find the corrections appropriate and have no further concerns. I accept the manuscript in its current form.
Sincerely yours,
Reviewer
Author Response
Many thanks for the fair review.
Round 3
Reviewer 1 Report
Comments and Suggestions for Authors
The authors corrected article , and now it is suitable for publication.